# Association between a Marine Healing Program and Metabolic Syndrome Components and Mental Health Indicators

**DOI:** 10.3390/medicina59071263

**Published:** 2023-07-06

**Authors:** Woo-Jin Byeon, Sung-Jae Lee, Tae-Gyu Khil, Ah-Young Jeong, Byoung-Duck Han, Min-Sung Sohn, Jae-Wook Choi, Yang-Hyun Kim

**Affiliations:** 1Department of Public Health, Graduate School, Korea University, 73 Goryeodae-ro, Seongbuk-gu, Seoul 02841, Republic of Korea; wj_9103@korea.ac.kr; 2Department of Integrative Medicine, College of Medicine, Korea University, 73 Goryeodae-ro, Seongbuk-gu, Seoul 02841, Republic of Korea; lee3676@korea.ac.kr (S.-J.L.); ktg0704@hanmail.net (T.-G.K.); honggilyue@naver.com (A.-Y.J.); 3Department of Family Medicine, Korea University Anam Hospital, Korea University College of Medicine, 73 Goryeodae-ro, Seongbuk-gu, Seoul 02841, Republic of Korea; hbdfm@korea.ac.kr; 4Department of Health and Medical Sciences, Cyber University of Korea, 161 Jeongneung-ro, Seongbuk-gu, Seoul 02708, Republic of Korea; minsinge@cuk.edu; 5Department of Preventive Medicine, College of Medicine, Korea University, 73 Goryeodae-ro, Seongbuk-gu, Seoul 02841, Republic of Korea

**Keywords:** metabolic syndrome, Korean, marine healing program, efficacy

## Abstract

*Background and Objectives*: Metabolic syndrome is a growing health concern globally, and its prevalence continues to increase. This study investigated whether a marine healing program could improve metabolic syndrome indicators and mental health in adults with a metabolic syndrome and those at risk of developing it. *Materials and Methods*: This study enrolled 30 participants who were assigned to either the experimental or control groups. The duration of the study was set at 4 weeks. Both groups received metabolic syndrome management education, and the experimental group additionally participated in two marine healing programs. Anthropometric indicators, biochemical indicators, and mental health indicators were collected before and after the intervention. *Results*: The findings indicate that the experimental group had significantly lower systolic blood pressure, triglycerides, and body weight, as well as higher levels of high-density lipoprotein (HDL-C) and uric acid. Mental health indicators (Hospital Anxiety and Depression Scale and quality of life measures) additionally showed improvement. Pre–post comparisons between the experimental group and the control group showed that the experimental group had significantly decreased by 1.05 kg in body weight, whereas the control group increased by 0.29 kg in body weight. In addition, HDL-C decreased by 0.91 mg/dL in the control group and increased by 3.7 mg/dL in the experimental group. *Conclusions*: Overall, these results suggest that marine healing programs could improve metabolic syndrome indicators such as body weight and HDL-C better than the control treatment.

## 1. Introduction

The metabolic syndrome (MetS), a cluster of cardiovascular risk factors, includes insulin resistance, abdominal obesity, dyslipidemia, and hypertension [1,2]. Urbanization, sedentary lifestyles, and excessive calorie intake all contribute to the rise in the prevalence of MetS globally, making it a significant public health concern on a global scale [3]. According to estimates, approximately 25% of the global population has MetS, which means that it affects more than 1 billion people [4]. The prevalence of metabolic syndrome in Korea has significantly grown from 21.6% in 2007 to 22.9% in 2018 [5], with the prevalence varying by age. Diabetes, cardiovascular disease, and cancer are more likely to strike persons with MetS than those without it [6,7,8].

To treat and prevent MetS, a change in lifestyle is required, including a balanced diet and consistent exercise. Various local governments in Korea are implementing MetS management projects intended to reduce the prevalence of MetS and prevent cardiovascular disease. One such project, Seoul Metabolic Syndrome Management (SMESY), has been in place in Seoul since 2011 and offers citizens personalized healthcare, nutrition, and exercise, including MetS screening, healthcare consultation, and aftercare. The SMESY project is based on the five risk factors used to diagnose metabolic syndrome [9]. Despite the implementation of various MetS management projects in South Korea, the prevalence of MetS continues to increase [5], indicating a need for alternative interventions to reduce the risk of MetS.

Recent studies have focused on nature therapy, which uses the natural environment to improve physical and mental health. Several studies found that decreased incidences of diabetes, all cause and cardiovascular mortality have been linked to green space [10]. In one study, nine middle-aged men diagnosed with prehypertension were assessed before and after undergoing forest therapy, and the findings revealed a substantial decrease in both diastolic and systolic blood pressure following the forest therapy intervention [11]. Patients with non-insulin-dependent diabetes mellitus in Japan saw significantly lower blood glucose levels after undergoing forest therapy [12]. Most of the attention and research on the health-promoting effects of nature therapy has focused on forest environments and forest therapy. However, only a few studies have examined marine environments and resources. In a study of eighteen patients with chronic tinnitus, an exposure to ocean sounds was administered as a therapeutic intervention, and it produced some improvement in the severity of tinnitus experienced by the patients [13]. Another study of twenty-five patients with degenerative arthritis of the knee used ocean resources as a therapeutic intervention, that resulted in some improvement in patient symptoms [14]. However, no research has yet explored the effectiveness of marine healing programs in treating and preventing MetS.

Therefore, this study evaluated the effectiveness of a marine healing program in improving MetS indicators and mental health among individuals with MetS. The research objectives were as follows: (i) to confirm the program’s effectiveness in improving the anthropometric indicators of MetS; (ii) to assess improvements in MetS biochemical indicators; and (iii) to evaluate changes in mental health indicators of MetS following the marine healing program.

## 2. Materials and Methods

### 2.1. Participants

Participants in this study were recruited based on the following selection criteria: (i) adult male and female, aged 30 to 80 years, (ii) diagnosed with MetS at a medical examination within the past year, or with at least 1 risk factor for MetS during the experimental period, and (iii) living in Taean-gun, Chungcheonbuk-do, South Korea, living inland rather than the coast. All the selected participants were already enrolled in a medical center in Taean-gun.

The criteria for defining MetS were based on the National Cholesterol Education Program Adult Treatment Panel III revision and the waist circumference (WC) criteria suggested by the Korean Society of Cardiometabolic Syndrome [5]: (i) WC is greater than 90 cm for males and 85 cm for females, (ii) triglycerides (TG) greater than 150 mg/dL, or taking medication for dyslipidemia, (iii) high-density lipoprotein cholesterol (HDL-C) of less than 40 mg/dL for males and less than 50 mg/dL for females, or taking medication for dyslipidemia, (iv) fasting blood glucose (FBG) greater than or equal to 100 mg/dL, or taking glucose-lowering drugs (or insulin), and (v) more than or equal to 85 mmHg during the diastolic blood pressure (DBP), or greater than or equal to 130 mmHg during the systolic blood pressure (SBP), or taking blood pressure-lowering medication. Participants with uncontrolled hypertension (≥160/100 mmHg) in the past 3 months, uncontrolled diabetes (HbA1c ≥ 9%), or uncontrolled dyslipidemia (LDL ≥ 190 mg/dL) were excluded.

Initially, we recruited fifteen participants for the experimental group and fifteen for the control group for the pre-test. However, after the first marine healing program, two participants were excluded because they met the exclusion criteria. In addition, three participants from the experimental group dropped out because they did not want to participate in the second program. Four participants from the control group participated only in the pre-test and did not participate in the post-test. Thus, for the final analysis, we used the pre-test and post-test results from the ten participants in the experimental group and eleven participants in the control group who participated in all the courses during the four-week study period.

### 2.2. Study Design

In this study, we used the arbitrary assignment of participants to the experimental and control groups in consideration of the research environment, clinical research conditions, and characteristics of marine healing research. Figure 1 provides the flow of participant enrollment. Those who could participate in both two-day and three-night marine healing programs were assigned to the experimental group. The study period was four weeks, during which the experimental group received the marine healing program twice, and the control group did not receive any program. The marine healing program was conducted openly without blinding because the subjects participated directly in the intervention. After the pre-test, both groups received MetS prevention and management education, and the experimental group was guided to maintain a healthy lifestyle throughout the study period. The control group was instructed to maintain a healthy lifestyle at home, without exposure to the marine healing program. The study ended with a post-test at Taean-gun Medical Center, which was conducted on an empty stomach. This study was conducted in adherence to the principles outlined in the Declaration of Helsinki and received ethical approval from the Institutional Review Board of Korea University Medical Center (IRB number: 2021AN0536).

### 2.3. Study Site

Table 1 provides the study design of the research. The experimental group in this study underwent two two-night, three-day marine healing programs at Cheongpodae Beach in Taean-gun, Chungcheongbuk-do, South Korea, during the second and fourth weeks of the experimental period. The selection of Cheongpodae Beach as the experimental site was based on the suitability of its beach and pine forest. The first marine healing program was conducted from 30 October to 1 November 2021, and the second program took place from 10 November to 12 November 2021. The average temperature during the first program was 12.6 °C, whereas during the second program, it was 6.63 °C.

### 2.4. Program Setup and Progress

Table 2 provides schedule for the marine healing program. The marine healing program considered the ocean’s environmental and resource characteristics and included a physical activity (PA) intervention, stress reduction intervention, and dietary intervention. All PA interventions took place in the marine environment and in pine forests near the coast. These interventions comprised pine forest phytoncide therapy, terrainkur, sea breeze aerosol breathing, and white sand Pilates. The stress reduction interventions were conducted between 19:00 and 20:00, after the completion of all the outdoor activities and before sleep.

The stress reduction intervention consisted of a peat heat pack intervention and aromatherapy. Peat is a soil, containing bioactive organic substances that have not been fully decomposed by microorganisms for decades or centuries in forests and coastal regions, and it is used medicinally in Europe to alleviate arthritis, lower back pain, and neck pain by bathing or using heat packs [15]. Studies have additionally demonstrated the benefits of using peat packs and saltwater foot baths to improve depression and anxiety [16]. Therefore, the aim of using peat heat packs in this study was to examine their effects on the mental health indicators in the experimental group. Sea breeze aerosols contain salts that are beneficial for the entire respiratory system and have stress-relieving effects. In France and Germany, sea breeze aerosols are recommended for patients with asthma and other bronchial and pulmonary conditions. Thus, we investigated the effect of breathing in sea breeze aerosols.

Terrainkur is an exercise method that uses coastal roads and white sandy beaches. Exercising on sandy beaches requires more energy than exercising in city centers or on flat, solid terrain. Therefore, this study investigated whether terrainkur improved anthropometric indicators of MetS. Nordic walking uses special sticks to make walking a full-body exercise that involves both the upper and lower body. It has been reported to have physical benefits, such as improving blood pressure, athletic performance, and maximal oxygen uptake [17]. Therefore, this study investigated improvements in anthropometric indicators after Nordic walking.

The dietary intervention provided a seaweed-based, low-salt diet for breakfast, lunch, and dinner. A detailed schedule of the marine healing program is provided in the table below.

### 2.5. Anthropometric Indicators

This study measured the following anthropometric indicators: height (cm), body weight (kg), WC (cm), skeletal muscle mass (SMM, kg), body fat mass (BFM, kg), body mass index (BMI, kg/m^2^), percent body fat (PBF, %), SBP (mmHg), and DBP (mmHg). Both pre-test and post-test measurements were conducted by a trained measurer, using the same body composition analyzers (manufacturer: InBody Co., Ltd., Seoul, Republic of Korea., model: InBody270) and sphygmomanometer (manufacturer: InBody Co., Ltd., model: BPBIO320S) at Taean-gun Medical Center to ensure the consistency and accuracy of the data.

### 2.6. Biochemical Indicators

FBG (mg/dL), aspartate aminotransaminase (AST, IU/L), alanine aminotransferase (ALT, IU/L), total cholesterol (mg/dL), TG (mg/dL), HDL-C (mg/dL), LDL-C (mg/dL), gamma-glutamyl transpeptidase (GGT, IU/L), uric acid (UAC, mg/dL), and HbA1c (%) were assessed as biochemical indicators. For both the pre-test and post-test, blood samples were collected at the Taean Medical Center and subsequently analyzed by the Seegene Medical Foundation.

### 2.7. Mental Health Indicators

Four survey instruments were used to assess the participants’ general mental health at the pre-test and post-test. First, the Korean version of the Hospital Anxiety and Depression Scale (HADS) was used to evaluate generalized anxiety and depression. The HADS is a well-established assessment, comprising 14 questions that can be completed by patients during their waiting time at the hospital [18]. The HADS was chosen because it is relatively easy and short and can assess patients’ anxiety and depression in primary care settings outside of psychiatry. A total of 7 of its items measure anxiety levels, and the other 7 items measure depression, with each item scored on a 4-point scale ranging from 0 (not at all) to 3 (very much so) and a total score of up to 21 points for anxiety and depression. Second, the EuroQol-five-dimensional questionnaire-three-level version (EQ-5D-3L) was used to measure the participants’ general quality of life (QOL). This tool comprises five questions related to mobility, self-care, usual activities, pain/discomfort, and anxiety/depression. All questions are scored on a scale from 1 (not at all impaired) to 3 (very impaired), and higher scores indicate worse QOL [19]. Third, to assess the effects of stress on health, the Korean version of the Brief Encounter Psychological Instrument (BEPSI-K) was used to evaluate the participants’ stress [20]. The BEPSI-K contains 5 questions, each scored on a scale from 0 (never) to 4 (always), and all scores are summed and divided by 5 to calculate an average score. Those scores are then divided into 3 groups, with a score of less than 1.8 indicating low stress, 1.8 to 2.8 indicating moderate stress, and 2.8 or more indicating high stress. Fourth, the Cognitive Stress Response Scale (CSRS) was used to further measure stress. The CSRS comprises 21 questions, each scored on a 5-point Likert scale ranging from 0 (not at all) to 4 (very much), with lower scores indicating lower levels of stress.

### 2.8. Statistical Analysis

The statistical analyses were performed as follows. First, an independent sample *t*-test was performed to determine the degree of equality in the pre-test results between the experimental and control groups. Second, normality tests were conducted for the pre-test and post-test variables, and those that satisfied normality were subjected to paired sample *t*-tests in both the experimental and control groups. Variables that did not satisfy normality were subjected to the non-parametric Wilcoxon signed rank test. Third, to compare the differences in test results between the two groups, independent sample *t*-tests were used for variables that showed normality, and non-parametric Mann–Whitney U tests were used for variables that did not show normality. All statistical analyses were performed in SPSS version 26.0 (IBM Corp., Armonk, NY, USA), with a *p*-value < 0.05 considered statistically significant.

## 3. Results

### 3.1. Demographic and Clinical Data

Information about the age, sex, pre-test anthropometric indicators, pre-test biochemical indicators, and pre-test mental health indicators of the participants is summarized in Table 3. The mean age of the experimental group was 68.10 years, which did not differ significantly from the mean age of the control group, which was 64.36 years (*p*-value = 0.318). Additionally, there were no differences between the 2 groups in the anthropometric indicators, biochemical indicators, or mental health indicators (*p*-values < 0.05). These findings suggest that the two groups were comparable in terms of their baseline characteristics.

Normality was tested using the Shapiro–Wilk test, and the results are presented in Appendix A. Based on the test results, variables that satisfied normality were analyzed using *t*-testing, and those that did not satisfy normality were analyzed using non-parametric testing. This approach ensures that appropriate statistical tests were used for each variable, taking into account the distribution of the data.

### 3.2. Comparison of Pre- and Post-Test Results in the Experimental Group: Normal Variables

Changes in the indicators from the experimental group after a 4-week study period were analyzed, using paired sample *t*-tests for the pre-test and post-test variables that met normality (Table 4). Body weight decreased significantly from 63.68 kg to 62.63 kg (*p*-value = 0.01). SBP decreased significantly from 152.70 mmHg to 134.20 mmHg (*p*-value = 0.008). HbA1c decreased from 5.70% to 5.56% (*p*-value = 0.050), and TG decreased significantly from 113.10 mg/dL to 79.50 mg/dL (*p*-value = 0.026). HDL-C increased significantly from 55.80 mg/dL to 59.50 mg/dL (p-value = 0.007), and UAC increased significantly from 4.88 mg/dL to 5.39 mg/dL (*p*-value = 0.009). The mental health indicators showed a significant decrease in the HADS score from 10.8 to 7.3 points (*p*-value = 0.048) and in the EuroQol EQ-5D-3L score from 7.3 to 5.9 points (*p*-value = 0.034). In addition, WC decreased from 92.250 cm to 90.90 cm, DBP decreased from 81.60 mmHg to 78.10 mmHg, and FBG decreased from 121.50 mg/dL to 117.80 mg/dL; however, those changes were not statistically significant.

### 3.3. Comparison of Pre- and Post-Test Results in the Experimental Group: Non-Normal Variables

Variables that did not meet normality were analyzed using the Wilcoxon signed rank sum test (Appendix A). Statistically significant changes were found in BFM (from 25.56 kg/m^2^ to 22.63 kg/m^2^; Z = −2.092, *p*-value = 0.036), GGT (from 20.20 IU/L to 16.30 IU/L; Z = −2.527, *p*-value = 0.012), and BEPSI-K (from 9.40 to 7.10; Z = −2.546, *p*-value = 0.011).

### 3.4. Comparison of Pre- and Post-Test Results in the Control Group: Normal Variables

To examine the changes in the control group after a 4-week study period, paired sample *t*-tests of the pre-test and post-test results were conducted for the variables that met normality (Table 5). The results revealed a statistically significant reduction in total cholesterol from 167.73 mg/dL to 156.55 mg/dL (*p*-value = 0.031) and a statistically significant decrease in the HADS score from 12.27 to 7.82 (*p*-value = 0.034). The increase found in HDL-C from 57.27 mg/dL to 56.36 mg/dL was not statistically significant.

### 3.5. Comparison of Pre- and Post-Test Results in the Control Group: Non-Normal Variables

For variables that did not meet normality, the Wilcoxon signed rank sum test was conducted on the pre-test and post-test results of the control group (Appendix A). The findings revealed a statistically significant decrease in SBP from 148.27 mmHg to 122.91 mmHg (Z = −2.397, *p*-value = 0.017), a statistically significant decrease in LDL-C from 96.73 mg/dL to 84.18 mg/dL (Z = −2.449, *p*-value = 0.014), and a statistically significant decrease in the EuroQol EQ-5D-3L score from 7.09 to 5.45 points (Z = −2.047, *p*-value = 0.041).

### 3.6. Comparison of Pre- and Post-Test Results between the Experimental and Control Groups

To investigate the efficacy of the marine healing program used in this study, independent sample *t*-tests were used to compare the differences between the pre-test and post-test results of the experimental and control groups for variables that satisfied the normality test (Table 6). The results of that analysis indicate that the body weight of the experimental group decreased by 1.05 kg, whereas that of the control group increased by 0.29 kg (t = 3.761, *p*-value < 0.001). Moreover, the experimental group exhibited a statistically significant increase of 3.7 mg/dL in HDL-C, whereas the control group displayed a decrease of 0.91 mg/dL (t = −2.168, *p*-value = 0.046).

For variables that did not meet normality assumptions, the Mann–Whitney U test was used to compare the pre-test and post-test differences between the experimental group and control group (Appendix A). The experimental group’s SMM, TG, and GGT decreased significantly compared with the control group.

## 4. Discussion

This study evaluated the effectiveness of a marine healing program in improving the anthropometric, biochemical, and mental health indicators of patients with or at risk of MetS. The experimental group exhibited a decrease in body weight, SBP, and TG and an increase in HDL-C and uric acid, as well as a decrease in the HADS and EuroQol EQ-5D-3L scores (all *p* < 0.05). Conversely, the control group displayed a decrease in total cholesterol and HADS scores (both *p* < 0.05) in the pre–post comparison. Body weight was reduced, and HDL-C improved significantly in the experimental group compared to the control group in the comparison of pre–post changes between the experimental and control groups (*p* = 0.001 and 0.046, respectively).

The experimental group’s body weight was considerably less than that of the control group, which did not take part in the marine healing program. This result is consistent with earlier research that has demonstrated that PA contributes to weight loss [21,22,23]. An imbalance between energy intake and expenditure frequently leads to weight fluctuation [24]. Reduced risk factors for type 2 diabetes and cardiovascular disease are linked to weight loss, and therefore, weight loss is recommended as a primary treatment for MetS [25]. The results of this study suggest that marine healing programs could be an effective way to promote regular PA and improve MetS.

The experimental group saw a decrease in TG and an increase in HDL-C. These results are consistent with previous studies that have demonstrated the beneficial effects of PA [26,27,28,29] and a diet based on seaweed [30]. Low levels of HDL-C are observed in 50% of patients with type 2 diabetes and are an independent risk factor for atherosclerotic cardiovascular disease [31]. Additionally, elevated TG are a risk factor for cardiovascular disease [32]. Therefore, reducing TG and increasing HDL-C are crucial for reducing the risk of MetS. However, the experimental group showed an increase in their UAC levels after the marine healing program. UAC was expected to decrease because high levels can cause gout; however, the consumption of seafood during the healing program appears to have increased UAC. Therefore, future studies must carefully design seaweed-based diets to prevent an increase in UAC.

The experimental group that underwent the marine healing program showed a decrease in GGT levels, compared with the control group. GGT is used as a diagnostic marker for liver dysfunction [33] and chronic alcohol abuse [34]. Multiple prospective studies have reported a positive correlation between increased GGT levels and mortality from all causes [35] and the development of MetS [36]. Thus, the decrease in GGT levels implies that the program tested here has the potential to improve liver function and, indirectly, improve MetS.

Although SBP, DBP, and mental health indicators improved numerically in the experimental group, those changes were not statistically significant compared with the control group. In future studies, it might be necessary to adjust the program to better target mental health indicators, such as by adding meditation or changing the aromas in the aromatherapy, because previous research has shown that a group stress intervention program using marine resources was effective in improving stress and some mental indicators [37]. Additionally, alternative measurement tools might be needed to better capture improved mental health.

Future studies with longer interventions, larger sample sizes, and randomized study designs are needed to complement these findings. Overall, the marine healing program might help to improve individual health and form a healthy culture when used in addition to conventional lifestyle interventions and drug treatment.

This study has several limitations that need to be considered. First, in this study, a small sample size was employed, which may have implications for the generalizability and reliability of the findings. Therefore, future research endeavors with larger and more diverse sample sizes are warranted to strengthen the external validity and enhance the robustness of the study’s outcomes. Second, the sample population was limited to MetS patients and those at risk of MetS living in Taean-gun, which could limit the generalizability of the results to other populations. Third, the study design was not a randomized controlled trial; therefore, selection bias cannot be ruled out. Fourth, the study was conducted for a short period, so it is unclear whether the observed health improvements will persist over time. Fifth, the analysis of the mental health indicators was limited by incomplete survey responses. Sixth, the program’s implementation was limited by the weather, which made it challenging to conduct the program in the late fall. Despite those limitations, this study contributes significantly to the field of marine healing, which is a natural healing intervention that has not been extensively studied. Additionally, this study has identified improvements in certain metabolic syndrome indicators, highlighting the potential effectiveness of the marine healing program.

## 5. Conclusions

In this study, the marine healing program produced a significant reduction in body weight and a significant increase in HDL-C in patients with MetS or at risk of MetS, compared with the control group. Therefore, further studies with larger sample sizes, longer intervention periods, and randomized study designs are warranted to strengthen these findings. Based on the current results, this marine healing program could be considered as an option for preventing and managing MetS, in addition to conventional lifestyle interventions and pharmacotherapy and for promoting individual health and a healthy culture.

## Figures and Tables

**Figure 1 medicina-59-01263-f001:**
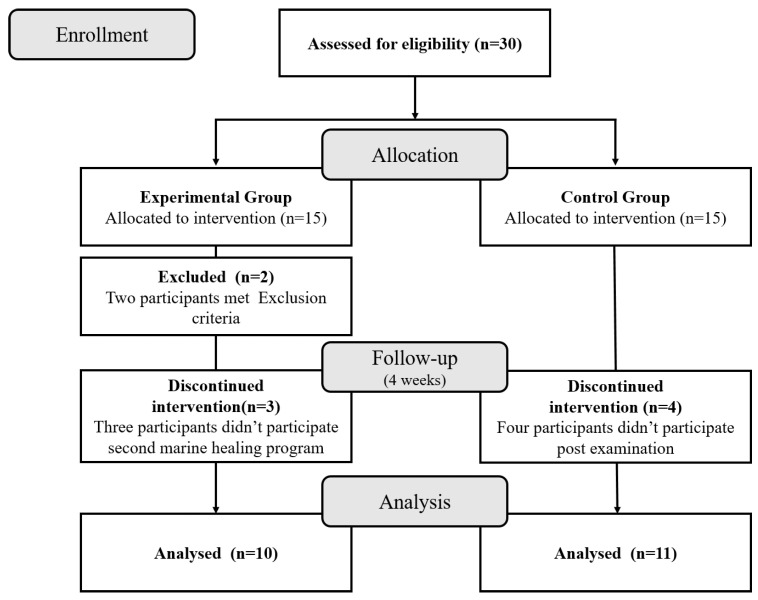
Flow chart of participant enrollment.

**Table 1 medicina-59-01263-t001:** Study design.

	Week 1	Week 2	Week 3	Week 4
Lifestyle intervention	Experimental group	Experimental group	Experimental group	Experimental group
	Control group	Control group	Control group	Control group
Marine healing program (2 nights, 3 days)		Experimental group		Experimental group

**Table 2 medicina-59-01263-t002:** Schedule for the marine healing program.

	1st			2nd		
Time	Oct 30	Oct 31	Nov 01	Nov 10	Nov 11	Nov 12
07:30~ 08:30		Breakfast (seaweed and low salt)	Breakfast (seaweed and low salt)		Breakfast (seaweed and low salt)	Post-test
09:00~ 10:30		PA in the pine forest	White sand Pilates		PA in the pine forest	
10:30~ 12:00		Terrainkur on white sandy beach	Terrainkur on white sandy beach		Terrainkur on white sandy beach	
12:00~ 13:00		Lunch (seaweed and low salt)	Lunch (seaweed and low salt)		Lunch (seaweed and low salt)	
13:00~ 14:00	Orientation	Sea breeze aerosol		Orientation	Sea breeze aerosol	
14:00~ 15:00	Medical examination			Medical examination		
15:00~ 16:00	Introduction to marine healing program	Tideland experience		Sea breeze aerosol	White sand Pilates	
16:00~ 17:00	PA in the pine forest			Nordic walking at beach	Terrainkur on white sandy beach	
17:00~ 18:00						
18:00~ 19:00	Dinner (seaweed and low salt)	Dinner (seaweed and low salt)		Dinner (seaweed and low salt)	Dinner (seaweed and low salt)	
19:00~ 20:00	Peat heat pack intervention	Aromatherapy		Peat heat pack intervention	Aromatherapy	

PA, physical activity.

**Table 3 medicina-59-01263-t003:** Demographic and clinical data.

Variable	Experimental		Control		*p*-Value
	Average	SD	Average	SD	
Sex F(M), n	6(4)		10(1)		0.120
Age (years)	68.10	7.16	64.36	9.28	0.318
Height (cm)	157.87	7.08	153.46	7.31	0.178
Body weight (kg)	63.68	7.42	64.56	8.75	0.809
WC (cm)	92.25	6.11	94.96	7.08	0.363
SMM (kg)	24.61	5.24	21.33	3.20	0.096
BFM (kg)	20.41	7.10	25.06	8.39	0.189
BMI (kg/m^2^)	25.56	2.68	27.45	4.03	0.227
PBF (%)	31.81	9.20	38.24	8.85	0.119
SBP (mmHg)	152.70	22.34	148.27	22.86	0.659
DBP (mmHg)	81.60	15.95	82.27	19.95	0.933
FBG (mg/dL)	121.50	23.80	107.91	12.97	0.116
HbA1c (%)	5.70	0.72	5.92	0.66	0.479
AST (IU/L)	25.30	5.87	27.18	4.26	0.408
ALT (IU/L)	20.50	9.10	26.91	10.23	0.147
Total cholesterol (mg/dL)	176.30	45.87	167.73	35.96	0.637
TG (mg/dL)	113.10	57.22	103.73	33.43	0.648
HDL-C (mg/dL)	55.80	13.13	57.27	14.49	0.811
LDL-C (mg/dL)	101.80	39.63	96.73	34.68	0.758
GGT (IU/L)	20.20	7.97	24.73	11.42	0.31
UAC (mg/dL)	4.88	0.92	4.55	0.93	0.431
HADS (score)	10.80	7.33	12.27	6.92	0.641
EQ-5D-3L (score)	7.30	1.34	7.09	1.81	0.769
BEPSI (score)	9.40	2.63	10.27	3.50	0.529
CSRS (score)	10.60	13.01	8.91	8.96	0.730

WC, waist circumference; SMM, skeletal muscle mass; BFM, body fat mass; BMI, body mass index; PBF, percent body fat; SBP, systolic blood pressure; DBP, diastolic blood pressure; FBG, fasting blood glucose; AST, aspartate aminotransaminase; ALT, alanine aminotransferase; TG, triglyceride; HDL-C, high density lipoprotein cholesterol; LDL-C, low density lipoprotein cholesterol; GGT, gamma-glutamyl transpeptidase; UAC, uric acid; HADS, Hospital Anxiety and Depression Scale; BEPSI, Brief Encounter Psychological Instrument; and CSRS, Cognitive Stress Response Scale.

**Table 4 medicina-59-01263-t004:** Comparison of pre- and post-test results in the experimental group.

Variable		Average	SD	t	df	*p*-Value
Height (cm)	pre	157.87	1.33	−0.74	9	0.479
	post	158.18				
Body weight (kg)	pre	63.68	1.03	3.23	9	0.010 **
	post	62.63				
WC (cm)	pre	92.25	4.18	1.02	9	0.333
	post	90.90				
SBP (mmHg)	pre	152.70	17.29	3.38	9	0.008 **
	post	134.20				
DBP (mmHg)	pre	81.60	13.71	0.81	9	0.440
	post	78.10				
FBG (mg/dL)	pre	121.50	13.38	0.87	9	0.405
	post	117.80				
HbA1c (%)	pre	5.70	0.2	2.26	9	0.050 *
	post	5.56				
AST (IU/L)	pre	25.30	3.28	−0.87	9	0.408
	post	26.20				
Total cholesterol (mg/dL)	pre	176.30	11.11	1.03	9	0.332
	post	172.70				
TG (mg/dL)	pre	113.10	39.84	2.67	9	0.026 *
	post	79.50				
HDL-C (mg/dL)	pre	55.80	3.34	−3.51	9	0.007 **
	post	59.50				
LDL-C (mg/dL)	pre	101.80	10.56	−0.03	9	0.977
	post	101.90				
UAC (mg/dL)	pre	4.88	0.48	−3.33	9	0.009 **
	post	5.39				
HADS (score)	pre	10.80	4.84	2.29	9	0.048 *
	post	7.30				
EQ-5D-3L (score)	pre	7.30	1.78	2.49	9	0.034 *
	post	5.90				

WC, waist circumference; SBP, systolic blood pressure; DBP, diastolic blood pressure; FBG, fasting blood glucose; AST, aspartate aminotransaminase; TG, triglyceride; HDL-C, high density lipoprotein cholesterol; LDL-C, low density lipoprotein cholesterol; UAC, uric acid; HADS, Hospital Anxiety and Depression Scale; * *p* < 0.05, ** *p* < 0.01.

**Table 5 medicina-59-01263-t005:** Comparison of pre- and post-test results in the control group.

Variable		Average	SD	t	df	*p*-Value
Body weight (kg)	pre	64.56	8.75	−1.73	10	0.115
	post	64.85	8.98			
SMM (kg)	pre	21.33	3.20	−1.70	10	0.120
	post	22.13	3.06			
BFM (kg)	pre	25.06	8.39	1.36	10	0.204
	post	23.97	8.08			
PBF (%)	pre	38.24	8.85	1.42	10	0.186
	post	36.38	8.51			
HbA1c (%)	pre	5.92	0.66	1.58	10	0.146
	post	5.84	0.64			
AST (IU/L)	pre	27.18	4.26	−0.99	10	0.344
	post	28.36	4.93			
ALT (IU/L)	pre	26.91	10.23	−1.17	10	0.267
	post	28.00	11.53			
Total cholesterol (mg/dL)	pre	167.73	35.96	2.50	10	0.031 *
	post	156.55	33.80			
HDL-C (mg/dL)	pre	57.27	14.49	0.49	10	0.633
	post	56.36	11.49			
UAC (mg/dL)	pre	4.55	0.93	−1.65	10	0.130
	post	4.74	0.93			
HADS (score)	pre	12.27	6.92	2.45	10	0.034 *
	post	7.82	5.74			
CSRS (score)	pre	8.91	8.96	−0.12	10	0.910
	post	9.27	9.47			

SMM, skeletal muscle mass; BFM, body fat mass; PBF, percent body fat; AST, aspartate aminotransaminase; ALT, alanine aminotransferase; HDL-C, high density lipoprotein cholesterol; UAC, uric-acid; HADS, Hospital Anxiety and Depression Scale; CSRS, Cognitive Stress Response Scale; * *p* < 0.05.

**Table 6 medicina-59-01263-t006:** Comparison of pre- and post-test results between the experimental and control groups.

Variable	Group	Pre-Test		Post-Test		Pre-Post	t	df	*p*-Value
		Mean	SD	Mean	SD				
Body weight (kg)	Exp.	63.68	7.42	62.63	6.73	1.05	3.761	19	0.001 ***
	Cont.	64.56	8.75	64.85	8.98	−0.29			
HbA1c (%)	Exp.	5.70	0.72	5.56	0.65	0.14	0.725	19	0.477
	Cont.	5.92	0.66	5.84	0.64	0.08			
AST (IU/L)	Exp.	25.30	5.87	26.20	4.42	−0.90	0.177	19	0.861
	Cont.	27.18	4.26	28.36	4.93	−1.18			
Total cholesterol (mg/dL)	Exp.	176.30	45.87	172.70	44.71	3.60	−1.314	19	0.204
	Cont.	167.73	35.96	156.55	33.80	11.18			
HDL-C (mg/dL)	Exp.	55.80	13.13	59.50	10.98	−3.70	−2.168	15.73	0.046 *
	Cont.	57.27	14.49	56.36	11.49	0.91			
UAC (mg/dL)	Exp.	4.88	0.92	5.39	0.84	−0.51	−1.764	19	0.094
	Cont.	4.55	0.93	4.74	0.93	−0.18			
HADS (score)	Exp.	10.80	7.33	7.30	6.31	3.50	−0.398	19	0.695
	Cont.	12.27	6.92	7.82	5.74	4.46			

AST, aspartate aminotransaminase; HDL-C, high density lipoprotein cholesterol; UAC, uric-acid; HADS, Hospital Anxiety and Depression Scale; * *p* < 0.05, *** *p* < 0.001.

## Data Availability

The data presented in this study are available on request from the corresponding author. The data are not publicly available due to participants’ private information.

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
