# Peer review of "Association between a Marine Healing Program and Metabolic Syndrome Components and Mental Health Indicators"

_medicina, 2023, doi:10.3390/medicina59071263_

Round 1

Reviewer 1 Report

The article is very actual and presented in a well-structured manner. The abstract of manuscript needs to be supplemented with data about duration of the follow-up. It is necessary to indicate how much the body weight has decreased and HDL-C has increased in the experimental and control groups.

Author Response

Dear Reviewer

Thank you for your valuable feedback on our manuscript. We appreciate your positive comments regarding the relevance and structure of the article.

Based on your suggestions, we have made the necessary revisions to the abstract. We have included data about the duration of the follow-up. Additionally, we have provided information on the extent of the decrease in body weight and increase in HDL-C for both the experimental and control groups.

We presented the duration of the of the study in the abstract and the results were as follows.

Line17: The duration of the study was set at 4 weeks.

We presented the results for weight and HDL-C in the abstract and the results were as follows.

Line23~26: Pre–post comparisons between the experimental group and the control group showed that the experimental group had significantly decreased 1.05 kg while the control group increased 0.29 kg in body weight. In addition, HDL-C decreased by 0.91 mg/dL in the control group and increased by 3.7 mg/dL in the experimental group.

We believe that these revisions have strengthened the manuscript by providing more comprehensive and specific results. We are grateful for your insightful suggestions, which have undoubtedly improved the overall quality of the article.

Thank you again for your time and valuable input. We look forward to hearing any further feedback you may have.

Best regards,

[Woo Jin Byeon]

Reviewer 2 Report

The manuscript is well-organized, but the sample size is very small. Moderate proofreading is required.

  • Some paragraphs and sentences are too long!

Avoid starting a sentence with an abbreviation, such as Line 37: MetS prevalence n Korea has

Line 134:   should be ….. included a Physical activity (PA) …..The second use for the entire manuscript should be PA, except for those in the initial of a sentence.

Line 190: Euro Qol–five‐dimensional questionnaire–three‐level version (EQ‐5D‐3L)

Moderate proofreading is required.

Author Response

Dear Reviewer

Thanks for pointing that out. We appreciate your positive comments about the organization of the article. In our paper, we have suggested on line 327 that future studies should be conducted with a larger sample size. We acknowledge your concerns regarding the small sample size, and we provide more detail on the limitation about sample size in the discussion and the results were as follows:

Line 330~334: First, in this study is the small sample size employed, which may have implications for the generalizability and reliability of the finding. So Future research endeavors with larger and more diverse sample sizes are warranted to strengthen the external validity and enhance the robustness of the study's outcomes.

Regarding your comment on proofreading, but we would like to clarify that we have already enlisted the assistance of a professional proofreader for this manuscript.

Based on your suggestions, we cleaned up a few sentences and the results were as follows:

Line58~60: Most of the attention and research on the health-promoting effects of nature therapy has focused on forest environments and forest therapy. However, with only a few studies examining marine environments and resources.

Line94~95: Initially, we recruited fifteen participants for the experimental group and fifteen for the control group for the pre-test. However, after the first marine healing program, two participants were excluded because they met the exclusion criteria. Also, three participants from the experimental group dropped out because they did not want to participate in the second program.

We appreciate your suggestion to avoid starting a sentence with an abbreviation. We will revise Line 37 to ensure that it reads smoothly without the abbreviation.

Line 37: The prevalence of metabolic syndrome (MetS) in ~

Furthermore, we will make the necessary adjustments throughout the manuscript to use "PA" instead of "Physical activity (PA)" except when it appears at the beginning of a sentence.

Line 134: physical activity (PA), Line 35, , Figure 3, Line 294, Line 299, Line 302: PA.

Lastly, we will address Line 190 by providing the full reference as follows: "Euro Qol–five‐dimensional questionnaire–three‐level version (EQ‐5D‐3L)."

We are grateful for your thorough review and constructive feedback, which will undoubtedly enhance the clarity and quality of our manuscript. If you have any further suggestions or concerns, please do not hesitate to let us know.

Thank you once again for your valuable input.

Best regards,

[Woo Jin Byeon]
